# A Molecular Phylogenetic Study of the Genus *Phedimus* for Tracing the Origin of “Tottori Fujita” Cultivars

**DOI:** 10.3390/plants9020254

**Published:** 2020-02-17

**Authors:** Sung Kyung Han, Tae Hoon Kim, Jung Sung Kim

**Affiliations:** 1Department of Forest Science, Chungbuk National University, Chungcheongbuk-do 28644, Korea; tjdrud0718@gmail.com; 2National Korea Forest Seed & Variety Center, Chungcheongbuk-do 27495, Korea; algae23@korea.kr

**Keywords:** *Phedimus takesimensis*, cultivar, origin, molecular phylogeny

## Abstract

It is very important to confirm and understand the genetic background of cultivated plants used in multiple applications. The genetic background is the history of crossing between maternal and paternal plants to generate a cultivated plant. If the plant in question was generated from a simple origin and not complicated crossing, we can easily confirm the history using a phylogenetic tree based on molecular data. This study was conducted to trace the origin of “Tottori Fujita 1gou” and “Tottori Fujita 2gou”, which are registered as cultivars originating from *Phedimus kamtschaticus*. To investigate the phylogenetic position of these cultivars, the backbone tree of the genus *Phedimus* needed to be further constructed because it retains inarticulate phylogenetic relationships among the wild species. We performed molecular phylogenetic analysis for *P. kamtschaticus*, *Phedimus takesimensis*, *Phedimus aizoon,* and *Phedimus middendorffianus*, which are assumed as the species of origin for “Tottori Fujita 1gou” and “Tottori Fujita 2gou”. The molecular phylogenetic tree based on the internal transcribed spacer (ITS) and *psb*A-*trn*H sequences showed the monophyly of the genus *Phedimus*, with *P. takesimensis* forming a single clade. However, *P. kamtschaticus* and *P. aizoon* were scattered in the tree. It was verified that “Tottori Fujita 1gou” and “Tottori Fujita 2gou” were embedded in a clade with *P. takesimensis* and not *P. kamtschaticus*. Therefore, origination from *P. takesimensis* was strongly supported. Based on these results, molecular phylogenetic analysis is suggested as a powerful tool for clearly tracing the origin of cultivated plants.

## 1. Introduction

Crassulaceae belong to the order Saxifragales of the core eudicots. It is also called the “stonecrop family” because it mostly comprises perennial herbaceous plants and many members have fleshy leaves. In this family, the Angiosperm Phylogeny Group (APG) IV system [1] currently contains three subfamilies, namely, Crassuloideae, Kalanchoideae, and Sempervivoideae, and it includes around 29–34 genera and 1400 species. Crassulaceae is distributed worldwide and has high species diversity in South Africa, Mexico, and mountainous areas in Asia. The family lives mostly in dry locations and is often found in saline areas. Crassulaceae is an important plant group that has been developed as horticultural cultivars for various morphological characteristics and vigorous growth in dry environments.

Regarding Crassulaceae classification, Berger [2] recognized 35 genera and 15,000 species in six subfamilies, but the largest genus, *Sedum* L., contains over 500 species with unclear phylogenetic relationships. To understand the taxonomic limitations of *Sedum*, numerous approaches have been undertaken by many researchers. Among them, Rafinesque [3] described an independent genus *Phedimus* Raf. from the genus *Sedum,* including the two species *Phedimus uniflorus* Raf. and *Phedimus stellatus* (L.) Raf., which have five-part calyx, unequal sepals longer than the petals, five equal petals, 10 stamens, and five ovaries. The author ‘t Hart [4] examined new combinations in *Phedimus* using phylogenetic analysis. They divided Crassulaceae into two subfamilies, namely, Crassuloideae and Sedoideae, and separated the subfamily Sedoideae into tribes Kalanchoeae and Sedeae. The Sedeae were again divided into subtribes Telephiinae and Sedinae, with Telephiinae including the genera *Rhodiola* L., *Hylotelephium* H. Ohba, and *Orostachys* Fisch. as well as *Meterostachys* Nakai and *Phedimus*, which were newly combined. Later, Ohba [5] divided *Phedimus* into subgenera *Phedimus* and *Aizoon* and established the current concept of the genus *Phedimus*, including the variants of *P. aizoon* and *Phedimus hsinganicus* (Y.C. Chu ex S.H. Fu and Y.H. Huang) H. Ohba, K.T. Fu, and B.M. Barthol. Since then, numerous molecular phylogenetic studies on Sedinae have been continuously conducted, and *Phedimus* has been separated from existing *Sedum* species. However, researchers have often continued to use a broad concept of intermixed “*Sedum*” and the newly defined “*Phedimus*” until now, even though the monophyly of *Phedimus* has been proven by molecular phylogenetic analysis [6,7,8]. Therefore, we have treated the genus *Phedimus* in accordance with the views of ‘t Hart [4] and Ohba [5] here.

For a long time, *Phedimus* has been primarily used to develop horticultural cultivars. The increasing of various developed and cultivated plants has led to the issue of their origination due to ambiguous information about their genetic origins. For example, *P. takesimensis* (Nakai) ‘t Hart is an endemic species to South Korea and was first introduced as “*Sedum takesimense* Nakai” at the UK Horticultural Society magazine “Sedum Society” in the early 1990s [9,10]. *P. takesimensis* was called the “evergreen Sedum” in this magazine because, unlike other plants in the Aizoon group that were widespread in Europe at the time (such as *Sedum aizoon* L., *Sedum kamtschaticum* Fisch., *Sedum middendorffianum* Maxim., *Sedum kurilense* Vorosch., and *S. kamtschaticum* subsp. *ellacombianum* (Praeger) R.T. Clausen), it retains green leaves even in the winter season. Because of these characteristics, *P. takesimensis* has been recognized as an important material for developing a horticultural plant based on its potential value, and it has entered the European market for greening and gardening. It has also been used as road landscaping plants since it was introduced to Japan in 2011 [11] as well as sold in Japanese markets in the form of seeds or young potted plants. In the case of the newly registered cultivars of “Tottori Fujita 1gou” and “Tottori Fujita 2gou”, the applicant reported *P. kamtschaticus* (Fisch.) ‘t Hart as the plant of origin to the National Korea Forest Seed and Variety Center in South Korea. However, this was controversial as the cultivated plants were morphologically closer to *P. takesimensis* than *P. kamtschaticus* (Figure 1). In the present study, we traced and clarified the definite origin of these cultivars using a molecular phylogenetic approach to overcome the limitations of discrimination via morphological characteristic comparisons. Since the controversy over the origin of these cultivars arose, no attempts have yet been made to resolve the issue using molecular phylogenetic analysis. Therefore, we suggest that it is possible to specifically trace the cultivar origin using molecular phylogenetic analysis in the case study of *Phedimus*.

## 2. Results

### 2.1. Sequence Variations within Phedimus

The nuclear ribosomal internal transcribed spacer (ITS) region (including ITS1, ITS2, 5.8S, and the partial 28S ribosomal gene) and the *psb*A-*trn*H spacer region of chloroplast DNA sequences of 85 individuals of the genus *Phedimus* were determined in this study.

Within the genus, the ITS region comprised 606–607 base pairs (bp), and the total aligned length was 608 bp. There were 471 constant sites (76.84% of all sites) and 115 parsimonious informative sites. In the aligned ITS region sequence, *P. kamtschaticus* and *P. aizoon* (L.) ‘t Hart showed similar nucleotide polymorphisms, although there was sequence variation within the same species. In *P. takesimensis*, excluding sample J6, the sequence was 607 bp in length, and “Tottori Fujita 1gou” and “Tottori Fujita 2gou” were also 607 bp. There were no species-specific variations in *P. takesimensis*; however, when comparing *P. takesimensis* sequences, they were divided into three types: α type (according to nucleotide substitution at 549 and 564 bp), β type (549 bp only), and γ type (564 bp only). “Tottori Fujita 1gou” and “Tottori Fujita 2gou” were both included in these three types (Figure 2). *P. middendorffianus* (Maxim) ‘t Hart, which showed a similar nucleotide sequence to *P. takesimensis*, had two species-specific nucleotide substitutions at 342 and 578bp, unlike other taxa.

In the *psb*A-*trn*H region, the length variation was 375–400 bp, and a small inversion was identified (Appendix A). There were 383 constant sites (90.12% of all sites) and 29 parsimonious informative sites in an aligned 413bp matrix. The shortest-length sequence of 375 bp had a 7 bp deletion of “GAATAGG” at 360–366 bp in the aligned sequences. Samples K8, K9, K10 (*P. takesimensis*), and K68 (“Tottori Fujita 1gou”), which had lengths of 377 bp, showed a 5bp deletion of “CATTT” at 200–204 bp. Only sample K48 (*P. aizoon* var. *floribundus*) had a length of 388bp due to a 6bp repeat sequence insertion of “CGGTAT” at 67–72 bp. The 7bp repeat sequence of “TTACTTA” was inserted at 166–172 bp in K38, K39, K40 (*P. kamtschaticus*), K42, K43, K47 (*P. aizoon*), and J13 (*P. aizoon* var. *floribundus*), which each had a length of 389 bp. K44 (*P. aizoon*) was 400 bp long due to a 25bp insertion at 162–186 bp and a 7bp deletion at 360–366 bp. A small inversion at 119–125 bp also occurred, but it was not a species-specific polymorphism. Species-specific nucleotide substitution appeared in *P. takesimensis* at 270 bp (C) (Appendix A). This same nucleotide substitution appeared in “Tottori Fujita 1gou” and “Tottori Fujita 2gou”, except for J6 and J7. Meanwhile, samples sharing the same deletion at 360–366 bp had common nucleotide substitutions at 261 (T), 266 (G), 270 (T), 318 (G), and 334 bp (G).

### 2.2. Origin and Phylogenetic Relationship of Cultivated Phedimus

Maximum likelihood (ML) trees of the ITS and *psb*A-*trn*H regions showed the monophyly of the genus *Phedimus* with 100% bootstrap support and 100% Shimodaira–Hasegawa (SH)-like approximate likelihood ratio test (SH-aLRT) support (Figure 3, Figure 4 and Figure 5). 

The tree length of the ITS region was 0.3632, and the sum of the internal branch lengths was 0.2592, accounting for 71.36% of the tree length. This was divided into two clades (Figure 3). *P. aizoon* and *P. kamtschaticus* formed Clade I with 75% bootstrap value and 90% SH-aLRT support. Clade II, comprising *P. takesimensis* and *P. middendorffianus*, was supported with 69% bootstrap value and 89% SH-aLRT support. *P. middendorffianus* formed a subclade within Clade II with 95% bootstrap value and 91.8% SH-aLRT support. *P. takesimensis* formed three subclades, which matched the three types of *P. takesimensis* sequence variation (α, β, and γ) of the ITS region. “Tottori Fujita 1gou”, “Tottori Fujita 2gou”, and *Phedimus* sp. were included in all three subclades of *P. takesimensis*. Regarding K31, K41, J6, J8, and J9, each showed a position that differed from their identification.

ML analysis for *psb*A-*trn*H was performed in two cases: corrected and uncorrected small inversion. The ML tree length was 0.1818, and the sum of the internal branch lengths was 0.0991, which was 71.36% of the tree length. The tree Baysian Information Criterion (BIC) score was 2128.3260 (Figure 4). The highly supported long branch of the tree (100% bootstrap value and 99.7% SH-aLRT support; taxa within the box in Figure 4) collapsed into the tree after correction of the inversed sequence (Figure 5). The ML tree length with corrected inversion was 0.1619, and the sum of the internal branch lengths was 0.0805, which was 49.70% of the tree length. The BIC score was 1943.4605. In the tree, only *P. takesimensis* composed a unique clade, unlike other species, with 80% bootstrap value and 86.6% SH-aLRT support. On the other hand, *P. kamtschaticus*, *P. aizoon*, and *P. middendorffianus* were scattered in the tree.

## 3. Discussion

Molecular phylogenetic study showed that the genus *Phedimus* separated from *Sedum* was monophyletic. Although there were several exceptions, closer relationships were found between *P. takesimensis* and *P. middendorffianus* as well as *P. kamtschaticus* and *P. aizoon* in the genus. Interestingly, three haplotypes were found in *P. takesimensis*, and all cultivars were embedded in those types. 

### 3.1. The Origin of Cultivar “Tottori Fujita”

The controversy surrounding the origin of the two *Phedimus* cultivars “Tottori Fujita 1gou” and “Tottori Fujita 2gou” arose due to the lack of an established correct phylogenetic relationship of *Phedimus*. Although the reliance of cultivar origin on external morphological characteristics tends to depend on the arguments of the applicant, there is a problem with the authenticity of the claims due to the absence of definite classification characteristics, such as morphological variations within the species of the genus *Phedimus*. 

The genus *Phedimus* was separated from *Sedum*, the largest genus in Crassulaceae, and recombined by ‘t Hart [4] and Ohba [5]. However, there were uncertain phylogenetic relationships among wild species, which caused confusion in applying species names and identifying taxa. Thus, we expected that the difficulty of tracing the origin of the registered cultivars would be resolved based on the analysis of the molecular phylogenetic relationships of *Phedimus*.

The results showed that the cultivars “Tottori Fujita 1gou” and “Tottori Fujita 2gou”, which were registered as new cultivars at the National Korea Forest Seed and Variety Center, formed a clade with *P. takesimensis*. The results also strongly indicated that the parental plant samples (marked as *Phedimus* sp. in data) submitted by the applicant as the species of origin *P. kamtschaticus* when registering these two new cultivars at the National Korea Forest Seed and Variety Center were also embedded in the *P. takesimensis* clade rather than *P. kamtschaticus*. Therefore, contrary to the information provided by the applicant, it was demonstrated that the cultivars “Tottori Fujita 1gou” and “Tottori Fujita 2gou” originated from *P. takesimensis*.

### 3.2. Application of Molecular Phylogenetic Analysis to Evaluate Misidentified Plants

We could also reidentify several *Phedimus* samples that were used in the present study and suspected of misidentification. For example, in the cases of J6 and J7, they were germinated and grown from purchased *P. takesimensis* seeds, which are currently sold at seed stores in Nagano Prefecture, Japan. However, unlike other *P. takesimensis* samples, which have a 1bp gap at 515 bp in the ITS region, J6 and J7 have an additional 1bp gap at 398 bp and a 1bp nucleotide substitution at 113 bp. This shows the same pattern as *P. kamtschaticus* and *P. aizoon*; however, not all sequence variances are perfectly consistent with these two species. Furthermore, samples K41, J8, and J9, which were identified as *P. kamtschaticus,* were exactly the same as the J6 and J7 sequences, and they formed a single clade in all trees. Therefore, it is reasonable to recognize them as *P. kamtschaticus* or *P. aizoon* rather than *P. takesimensis.* Accordingly, *Phedimus* seeds released on the market should be handled carefully due to the probability of misidentification to avoid confusion.

### 3.3. Hybridization Leading to Discordance on the Phylogenetic Tree

Hybridization is an important mechanism that obtains new genotypes through a combination of different genomes. And then, the plant species is generally diversified into other species that are evolutionarily competitive [12]. Some plants are also generated by selective breeding that suits useful values, such as building a new lineage that maximizes the specific character manifestation [13]. In natural conditions, when an interspecific hybridization has been occurred, it may be used to recognize a new species and give a caution to carefully define the boundary of species [14]. Therefore, researchers have investigated hybridization in wild species and cultivars by morphological, physiological, and genetic approaches [15,16,17].

Hybrids originating from wild species of the genus *Phedimus* have been reported in Korean taxa [18,19]. In addition, various hybrid plants have been reported in the subspecies *P. aizoon,* and it has caused a lot of confusion in the recognition of Japanese species [20]. This is particularly the case for *P. kamtschaticus* and *P. aizoon,* which are generally identified by their leaf type. The boundary of species is still unclear because there is no apparent morphological difference between them, and interspecific hybrids have been reported.

The present study also showed that these two species were not distinguishable from each other in the phylogenetic tree. Clade I containing *P. kamtschaticus* and *P. aizoon* and the clade of *P. middendorffinus* were clearly recognizable in the ITS tree (Figure 3). However, all three species were scattered in the *psb*A-*trn*H tree (Figure 5). It is assumed that hybridization has occurred among species in *Phdimus* except for *P. takesimnesis*.

The K52 and K54 samples (*P. middendorffianus*) formed a subclade with another *P. middendorffianus* in the ITS tree. However, these samples were separated from them in the *psb*A-*trn*H tree and formed one subclade with several *P. kamtschaticus* (Figure 3 and Figure 5). This finding made us speculate that the K52 and K54 samples had different maternal origin than the rest of the *P. middendorffianus* samples.

In the case of sample K31 (*P. kamtschaticus*), a hybrid origin was also suspected because it showed a similar pattern to *P. takesimensis* in the ITS region but a similar tendency to *P. kamtschaticus* in the *psb*A-*trn*H region.

Meanwhile, the J10 sample (*P. aizoon*) was included in Clade I of the ITS tree, but in the case of the *psb*A-*trn*H tree, the J10 sample was separate from the clade that included all the *Phedimus* samples (Figure 3 and Figure 5). Besides, a unique sequence variation (A) was found at 337 bp position of the *psb*A-*trn*H sequence of J10. This suggests the possibility of de novo mutation in the J10 sample.

### 3.4. Molecular Marker Development for Further Study

In addition, we found the species-specific variations for *P. middendorffianus* at 342 bp in the ITS region (Figure 2) and at 275 bp in the *psb*A-*trn*H region for *P. takesimensis* (Appendix A). It is anticipated that these polymorphic sequence sites may be used to develop molecular markers to identify the two species among the different *Phedimus* species. 

In the horticultural and forestry industries, all countries encourage and promote development and sale based on cooperation with the International Union for the Protection of New Varieties of Plants (UPOV). However, the importance of providing accurate information for cultivated plants has often been ignored, including the history of their generation and genetic background to evaluate the characteristics of plants in the market. In fact, in most cases, a plant can be accepted as a new cultivar if it maintains just one unique characteristic compared to the plant of origin. To establish a system to protect plant cultivars, the genetic background must be provided along with the morphological features in the future. Based on the results of this study, it is suggested that the technique of molecular phylogenetic analysis be used to trace the definite origin of cultivars and to define the phylogenetic boundary between species. Molecular phylogenetic analysis can be used as a powerful tool to produce a genetic backbone tree for tracking the origin of cultivars.

## 4. Materials and Methods

### 4.1. Plant Materials and Sampling

In total, 85 individuals (69 from Korea and 16 from Japan) were used in this study, including the “Tottori Fujita 1gou” and “Tottori Fujita 2gou” cultivars, which were provided from the National Korea Forest Seed and Variety Center, and four species of *Phedimus*, which occur in wild habitats in Korea and Japan (Figure 1). In addition, the sequence data of the related taxa, i.e., *Pseudosedum* and *Rhodiola* [21,22,23], were downloaded from the National Center for Biotechnology Information (NCBI) database (Table 1).

### 4.2. DNA Extraction, PCR Amplification, and Sequencing

Total genomic DNA was extracted from silica-gel-dried leaf tissues using a modified CTAB method [24] or the DNeasy Plant Mini Kit (QIAGEN, Germany) following the manufacturer’s instructions. The quantity and quality of the extracted DNA were analyzed using 1% agarose gel electrophoresis with 1X Tri-acetate-EDTA (TAE) buffer and a spectrophotometer (Thermo Scientific™ NanoDrop 2000, Thermo Fisher Scientific, Waltham, MA, USA). We amplified the ITS region (rDNA) and the *psb*A-*trn*H region (cpDNA). To amplify the target regions, we used primer pairs of ITS1 and ITS4 [25] and of *psb*A and *trn*H [26] (Table 2). The PCR reaction for the ITS region was initialized at 95 °C for 1 min, followed by 35 cycles of denaturation at 95 °C for 30 s, annealing at 51 °C for 30 s, and extension at 72 °C for 1 min; the final extension step was performed at 72 °C for 5 min. For the *psb*A-*trn*H region, PCR was initialized at 95 °C for 1 min, followed by 35 cycles of denaturation at 95 °C for 30 s, annealing at 55 °C for 30 s, and extension at 72 °C for 1 min; the final extension step was performed at 72 °C for 5 min. The PCR products were run on 1% agarose gels in 1X TAE buffer and purified using the Expin PCR SV Mini Kit (GeneAll, Seoul, South Korea). Sequencing was carried out by the 3730XL Automated DNA Sequencing System (Applied Biosystems, Foster City, CA, USA).

### 4.3. Molecular Phylogenetic Analyses of ITS and Plastid DNA Regions

The nuclear ITS region sequence and plastid noncoding regions, *psb*A-*trn*H, were used to compose the phylogeny. Geneious 7.1.9 [27] was used to assemble the DNA sequences resulting from the PCR products. Sequences were initially aligned with the MAFFT v7.017 alignment program [28] in Geneious 7.1.9 using the default parameter values. Then, all sequences were manually checked, and amendments were directly made.

Phylogenetic analysis was performed using maximum likelihood methods with W-IQ-TREE 1.6.11 [29], and gaps were treated as missing data. The best evolutionary substitution model identified by the Bayesian information criterion [30] was selected at TIM3e + G4 for the ITS region, F81 + F + I for the *psb*A-*trn*H region after correcting for the small inversion, and F81 + F + G4 for the *psb*A-*trn*H region with uncorrected inversion using W-IQ-TREE 1.6.11. The supporting value for each clade was estimated from 1000 bootstrap replicates [31], and we performed 1000 replications for the SH-aLRT as the branch test [32].

## Figures and Tables

**Figure 1 plants-09-00254-f001:**
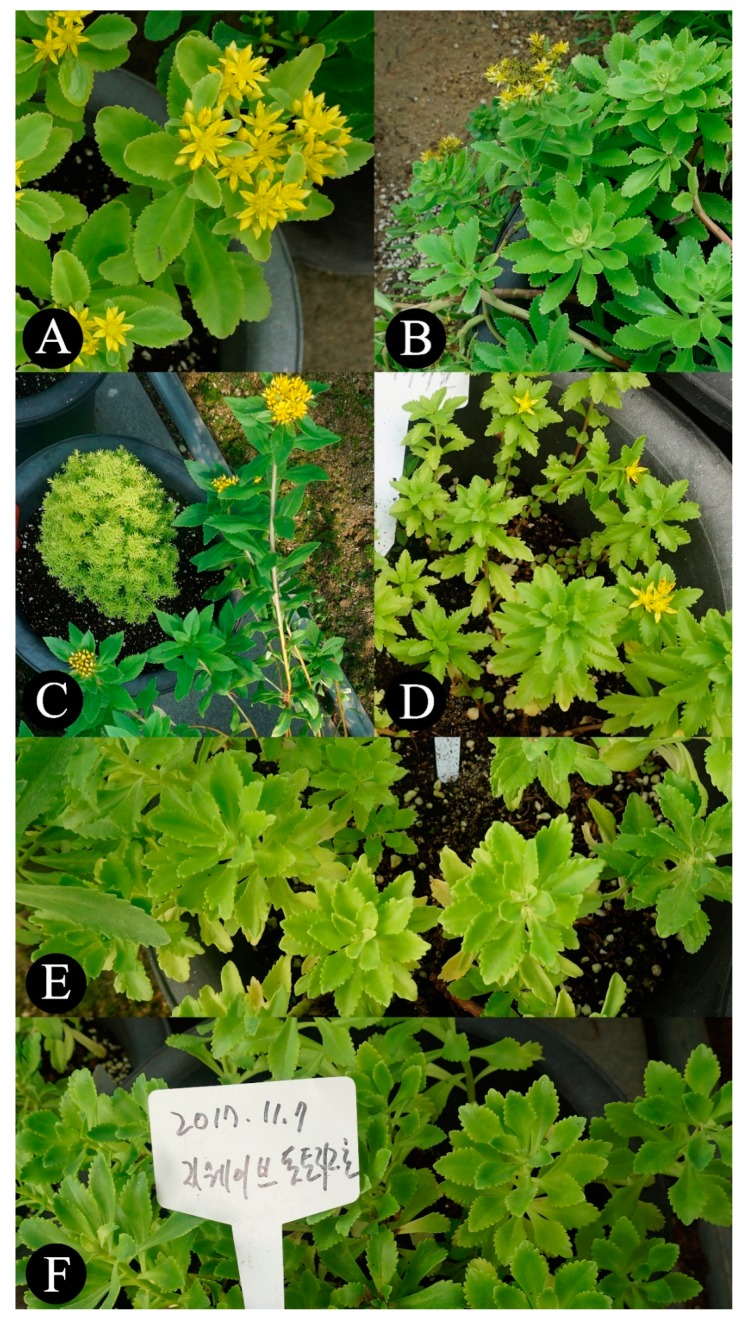
Plant materials of the genus *Phedimus* used for analysis. (**A**) *Phedimus kamtschaticus*, (**B**) *Phedimus takesimensis*, (**C**) *Phedimus aizoon*, (**D**) *Phedimus middendorffianus*, (**E**) “Tottori Fujita 1gou”, and (**F**) “Tottori Fujita 2gou”.

**Figure 2 plants-09-00254-f002:**
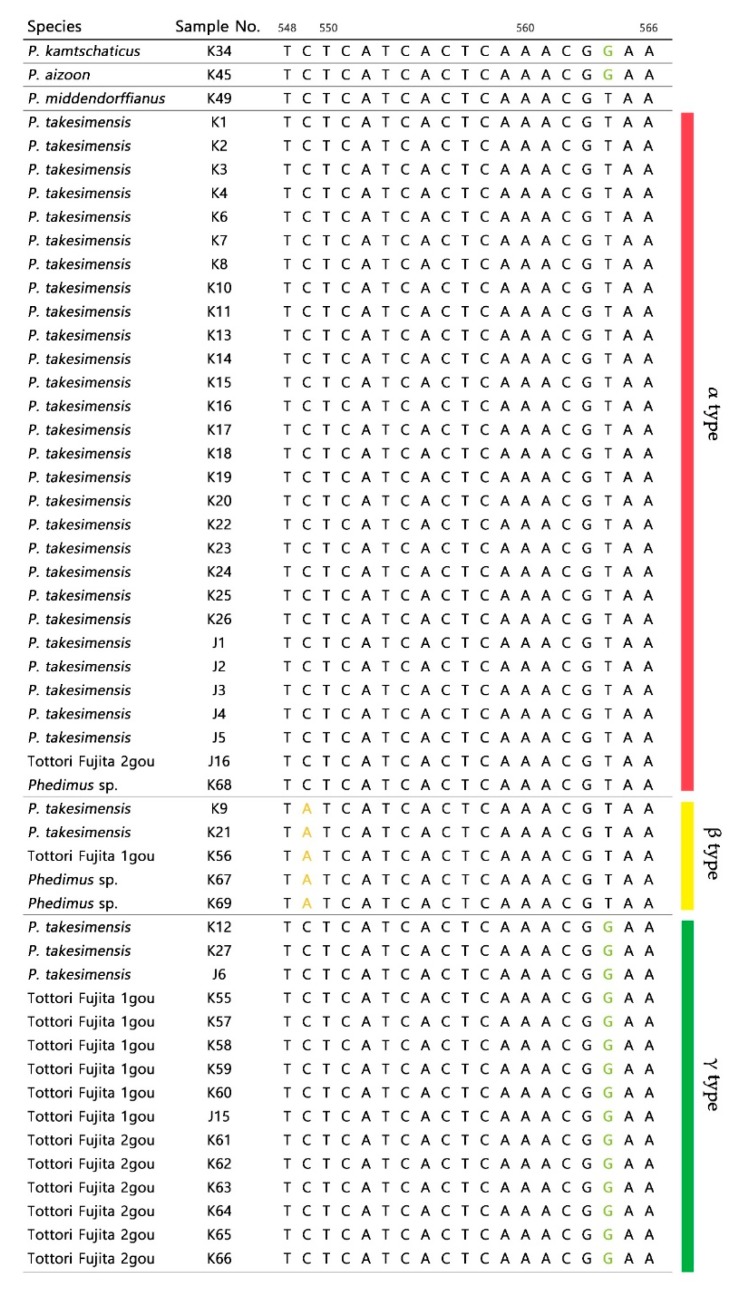
Three types based on nucleotide substitution in internal transcribed spacer (ITS) sequences of *P. takesimensis*, cultivar “Tottori Fujita 1gou” and “Tottori Fujita 2gou”.

**Figure 3 plants-09-00254-f003:**
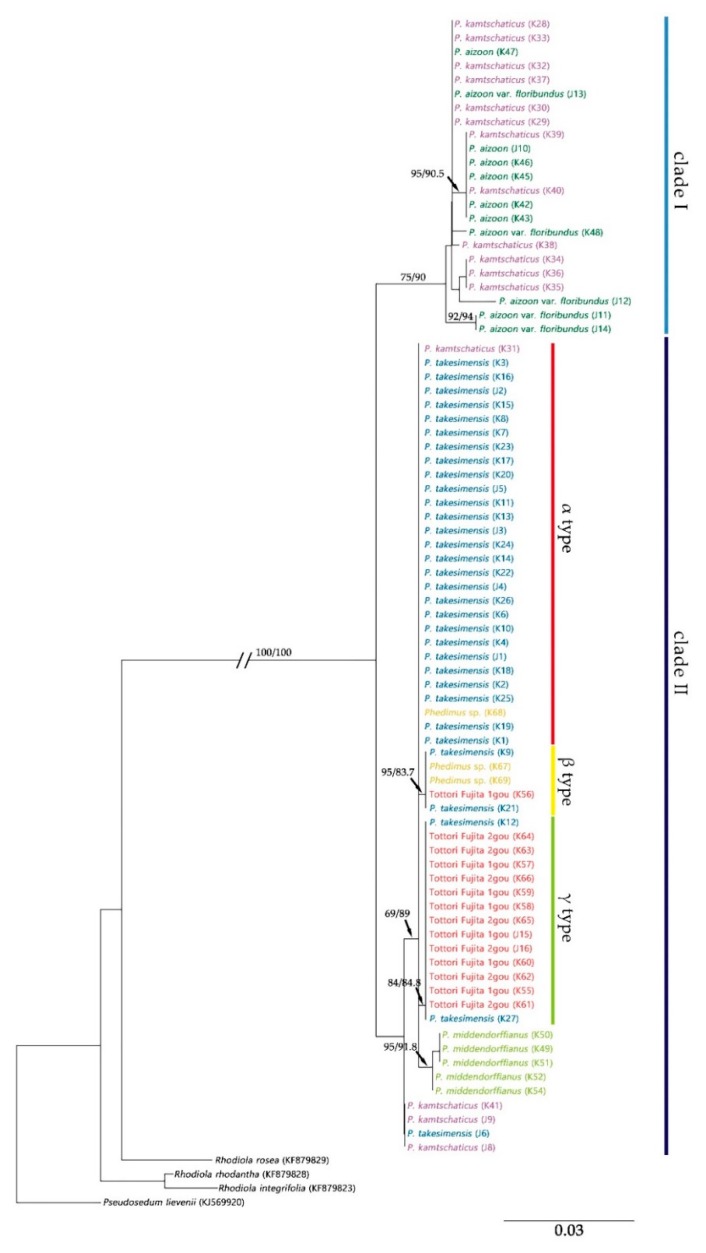
Maximum likelihood (ML) tree of the genus *Phedimus* based on ITS sequence. Numbers on the branches indicate the bootstrap values and Shimodaira–Hasegawa (SH)-like approximate likelihood ratio test (SH-aLRT) support.

**Figure 4 plants-09-00254-f004:**
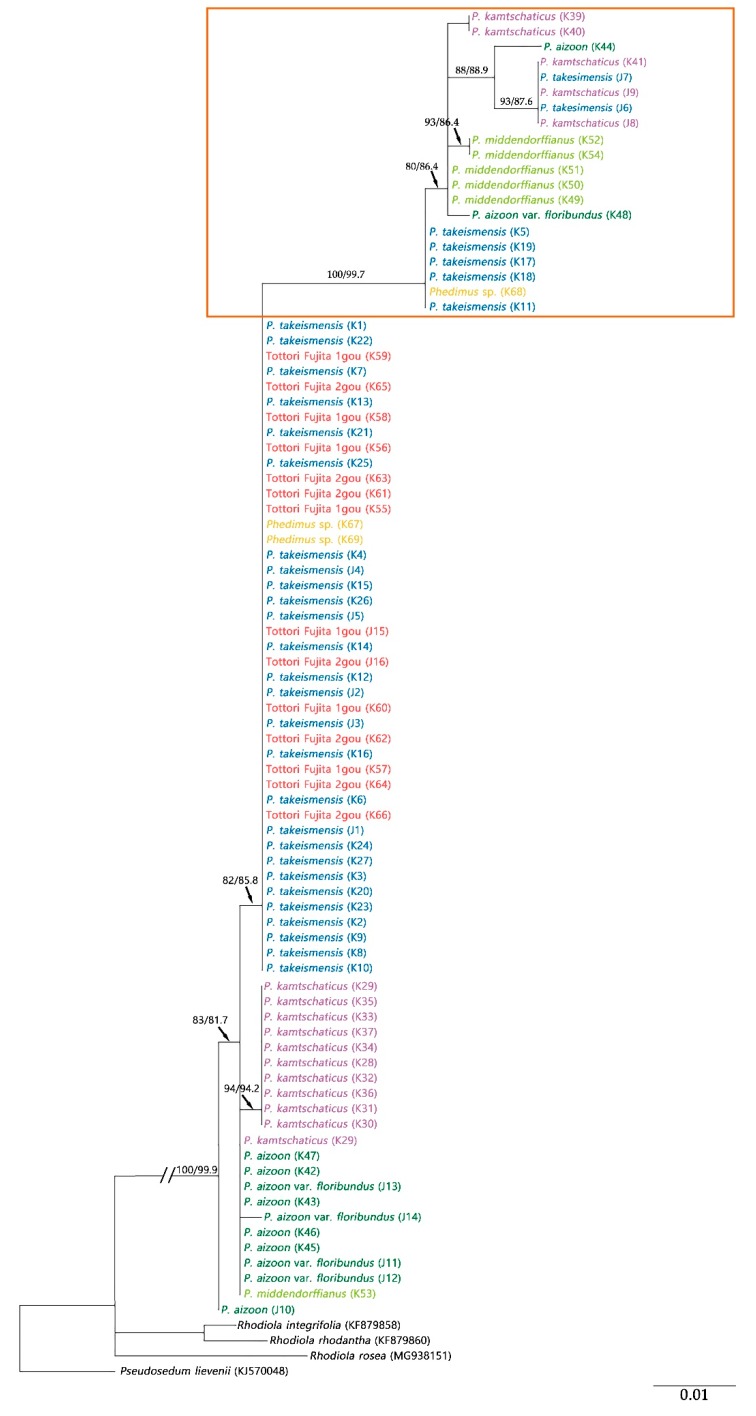
ML tree of the genus *Phedimus* using *psb*A-*trn*H sequence without correcting for the inversion. Numbers on branches indicate the bootstrap values and SH-aLRT support.

**Figure 5 plants-09-00254-f005:**
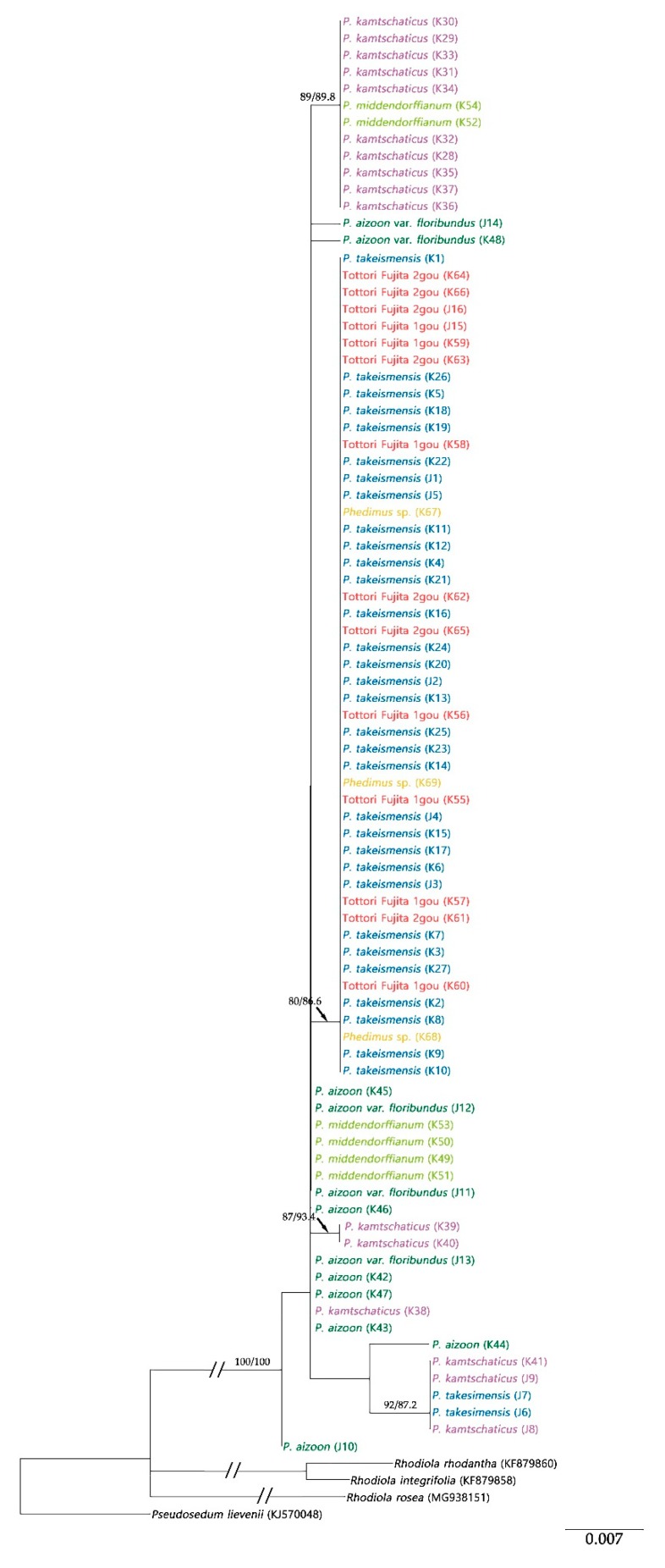
ML tree of the genus *Phedimus* using *psb*A-*trn*H sequence after correcting for the inversion. Numbers on branches indicate the bootstrap values and SH-aLRT support.

**Table 1 plants-09-00254-t001:** List of plant materials with accession numbers deposited in the National Center for Biotechnology Information (NCBI) database.

Sample ID	Species	Collection Information of Locality or Reference	Acc. No.	
			ITS	*psb*A-*trn*H
K1	*Phedimus takesimensis*	Ulleung-gun, Gyeongsangbuk-do, Korea	MN908990	MN935682
K2	*Phedimus takesimensis*	Ulleung-gun, Gyeongsangbuk-do, Korea	MN908991	MN935683
K3	*Phedimus takesimensis*	Ulleung-gun, Gyeongsangbuk-do, Korea	MN908992	MN935684
K4	*Phedimus takesimensis*	Ulleung-gun, Gyeongsangbuk-do, Korea	MN908993	MN935685
K5	*Phedimus takesimensis*	Ulleung-gun, Gyeongsangbuk-do, Korea	-	MN935686
K6	*Phedimus takesimensis*	Ulleung-gun, Gyeongsangbuk-do, Korea	MN908994	MN935687
K7	*Phedimus takesimensis*	Ulleung-gun, Gyeongsangbuk-do, Korea	MN908995	MN935688
K8	*Phedimus takesimensis*	collectcollected from the natural locality	MN908996	MN935689
K9	*Phedimus takesimensis*	collected from the natural locality (not specified)	MN908997	MN935690
K10	*Phedimus takesimensis*	collected from the natural locality (not specified)	MN908998	MN935691
K11	*Phedimus takesimensis*	Ulleung-gun, Gyeongsangbuk-do, Korea	MN908999	MN935692
K12	*Phedimus takesimensis*	Ulleung-gun, Gyeongsangbuk-do, Korea	MN909000	MN935693
K13	*Phedimus takesimensis*	Ulleung-gun, Gyeongsangbuk-do, Korea	MN909001	MN935694
K14	*Phedimus takesimensis*	Ulleung-gun, Gyeongsangbuk-do, Korea	MN909002	MN935695
K15	*Phedimus takesimensis*	Ulleung-gun, Gyeongsangbuk-do, Korea	MN909003	MN935696
K16	*Phedimus takesimensis*	Ulleung-gun, Gyeongsangbuk-do, Korea	MN909004	MN935697
K17	*Phedimus takesimensis*	Dokdo-ri, Ulleung-gun, Gyeongsangbuk-do, Korea	MN909005	MN935698
K18	*Phedimus takesimensis*	Dokdo-ri, Ulleung-gun, Gyeongsangbuk-do, Korea	MN909006	MN935699
K19	*Phedimus takesimensis*	Dokdo-ri, Ulleung-gun, Gyeongsangbuk-do, Korea	MN909007	MN935700
K20	*Phedimus takesimensis*	collected from the natural locality (not specified)	MN909008	MN935701
K21	*Phedimus takesimensis*	collected from the natural locality (not specified)	MN909009	MN935702
K22	*Phedimus takesimensis*	collected from the natural locality (not specified)	MN909010	MN935703
K23	*Phedimus takesimensis*	Dokdo-ri, Ulleung-gun, Gyeongsangbuk-do, Korea	MN909011	MN935704
K24	*Phedimus takesimensis*	Dokdo-ri, Ulleung-gun, Gyeongsangbuk-do, Korea	MN909012	MN935705
K25	*Phedimus takesimensis*	Dokdo-ri, Ulleung-gun, Gyeongsangbuk-do, Korea	MN909013	MN935706
K26	*Phedimus takesimensis*	Ulleung-gun, Gyeongsangbuk-do, Korea	MN909014	MN935707
K27	*Phedimus takesimensis*	Ulleung-gun, Gyeongsangbuk-do, Korea	MN909015	MN935708
K28	*Phedimus kamtschaticus*	Pohang-si, Gyeongsangbuk-do, Korea	MN908958	MN935733
K29	*Phedimus kamtschaticus*	Goseong-gun, Gangwon-do, Korea	MN908959	MN935734
K30	*Phedimus kamtschaticus*	Bonghwa-gun, Gyeongsangbuk-do, Korea	MN908960	MN935735
K31^a^	*Phedimus kamtschaticus*	Pocheon-si, Gyeonggi-do, Korea	MN908961	MN935736
K32	*Phedimus kamtschaticus*	Pocheon-si, Gyeonggi-do, Korea	MN908962	MN935737
K33	*Phedimus kamtschaticus*	Pocheon-si, Gyeonggi-do, Korea	MN908963	MN935738
K34	*Phedimus kamtschaticus*	Seocho-gu, Seoul, Korea	MN908964	MN935739
K35	*Phedimus kamtschaticus*	Seocho-gu, Seoul, Korea	MN908965	MN935740
K36	*Phedimus kamtschaticus*	Seocho-gu, Seoul, Korea	MN908966	MN935741
K37	*Phedimus kamtschaticus*	collected from the natural locality (not specified)	MN908967	MN935742
K38	*Phedimus kamtschaticus*	Pohang-si, Gyeongsangbuk-do, Korea	MN908968	MN935743
K39	*Phedimus kamtschaticus*	Pohang-si, Gyeongsangbuk-do, Korea	MN908969	MN935744
K40	*Phedimus kamtschaticus*	Pohang-si, Gyeongsangbuk-do, Korea	MN908970	MN935745
K41^b^	*Phedimus kamtschaticus*	Ulleung-gun, Gyeongsangbuk-do, Korea	MN908971	MN935748
K42	*Phedimus aizoon*	collected from the natural locality (not specified)	MN908974	MN935749
K43	*Phedimus aizoon*	collected from the natural locality (not specified)	MN908975	MN935750
K44	*Phedimus aizoon*	collected from the natural locality (not specified)	-	MN935751
K45	*Phedimus aizoon*	farm, Pyeongchang-gun, Gangwon-do, Korea	MN908976	MN935752
K46	*Phedimus aizoon*	farm, Pyeongchang-gun, Gangwon-do, Korea	MN908977	MN935753
K47	*Phedimus aizoon*	farm, Pyeongchang-gun, Gangwon-do, Korea	MN908978	MN935754
K48	*Phedimus aizoon* var. *florivundus*	Gyeongju-si, Gyeongsangnam-do, Korea	MN908980	MN935758
K49	*Phedimus middendorffianus*	collected from the natural locality (not specified)	MN908985	MN935761
K50	*Phedimus middendorffianus*	collected from the natural locality (not specified)	MN908986	MN935762
K51	*Phedimus middendorffianus*	collected from the natural locality (not specified)	MN908987	MN935763
K52	*Phedimus middendorffianus*	farm, Uiwang-si, Gyeonggi-do, Korea	MN908988	MN935764
K53	*Phedimus middendorffianus*	farm, Uiwang-si, Gyeonggi-do, Korea	-	MN935765
K54	*Phedimus middendorffianus*	farm, Uiwang-si, Gyeonggi-do, Korea	MN908989	MN935766
K55	“Tottori Fujita 1gou”	generated and submitted by applicant	MN909025	MN935716
K56	“Tottori Fujita 1gou”	generated and submitted by applicant	MN909026	MN935717
K57	“Tottori Fujita 1gou”	generated and submitted by applicant	MN909027	MN935718
K58	“Tottori Fujita 1gou”	generated and submitted by applicant	MN909028	MN935719
K59	“Tottori Fujita 1gou”	generated and submitted by applicant	MN909029	MN935720
K60	“Tottori Fujita 1gou”	generated and submitted by applicant	MN909030	MN935721
K61	“Tottori Fujita 2gou”	generated and submitted by applicant	MN909032	MN935723
K62	“Tottori Fujita 2gou”	generated and submitted by applicant	MN909033	MN935724
K63	“Tottori Fujita 2gou”	generated and submitted by applicant	MN909034	MN935725
K64	“Tottori Fujita 2gou”	generated and submitted by applicant	MN909035	MN935726
K65	“Tottori Fujita 2gou”	generated and submitted by applicant	MN909036	MN935727
K66	“Tottori Fujita 2gou”	generated and submitted by applicant	MN909037	MN935728
K67	*Phediums* sp.	generated and submitted by applicant	MN909023	MN935730
K68	*Phediums* sp.	generated and submitted by applicant	MN909022	MN935731
K69	*Phediums* sp.	generated and submitted by applicant	MN909024	MN935732
J1	*Phedimus takesimensis*	Seed store, Nagano Pref., Japan	MN909016	MN935709
J2	*Phedimus takesimensis*	Seed store, Nagano Pref., Japan	MN909017	MN935710
J3	*Phedimus takesimensis*	Seed store, Nagano Pref., Japan	MN909018	MN935711
J4	*Phedimus takesimensis*	Seed store, Nagano Pref., Japan	MN909019	MN935712
J5	*Phedimus takesimensis*	Seed store, Nagano Pref., Japan	MN909020	MN935713
J6 ^b^	*Phedimus takesimensis*	Seed store, Nagano Pref., Japan	MN909021	MN935714
J7 ^b^	*Phedimus takesimensis*	Seed store, Nagano Pref., Japan	-	MN935715
J8 ^b^	*Phedimus kamtschaticus*	Hokkaido, Obihiro-si, Japan	MN908972	MN935746
J9 ^b^	*Phedimus kamtschaticus*	Hokkaido, Memuro-cho, Japan	MN908973	MN935747
J10	*Phedimus aizoon*	Nagano Pref., Japan	MN908979	MN935755
J11	*Phedimus aizoon* var. *floribundus*	Nigata Pref. Kasiwazaki-si, Japan	MN908981	MN935756
J12	*Phedimus aizoon* var. *floribundus*	Kagawa Pref. Kankakei, Japan	MN908982	MN935757
J13	*Phedimus aizoon* var. *floribundus*	Seed store, Osaka Pref., Japan	MN908983	MN935759
J14	*Phedimus aizoon* var. *floribundus*	Nigata Pref. Sado-si, Japan	MN908984	MN935760
J15	“Tottori Fujita 1gou”	Tottori Pref. Iwami-cho, Japan	MN909031	MN935722
J16	“Tottori Fujita 2gou”	Tottori Pref. Iwami-cho, Japan	MN909038	MN935729
**Outgroup**				
	*Pseudosedum lievenii*	Zhang, J.Q. et al., 2014 [21]	KJ569920	KJ570048
	*Rhodiola integrifolia*	Guest, H.J. et al., 2014 [22]	KF879823	KF879860
	*Rhodiola rhodantha*	Guest, H.J. et al., 2014 [22]	KF879828	KF879860
	*Rhodiola rosea*	Guest, H.J. et al., 2014 [22], Gyorgy, Z., et al., 2018 [23]	KF879829	MG938151

a: plant material suspected of a hybrid origin, b: plant materials suspected of misidentification.

**Table 2 plants-09-00254-t002:** Primer information applied for the present study.

Region	Primer	Sequence (5′→ 3′)	Reference
ITS (rDNA)	ITS1	TCCGTAGGTGAACCTGCGG	White et al. [25]
	ITS4	TCCTCCGCTTATTGATATGC	White et al. [25]
*psb*A-*trn*H (cpDNA)	psbA3_f	GTTATGCATGAACGTAATGCTC	CBOL Plant Working Group [26]
	trnHf_05	CGCGCATGGTGGATTCACAATCC	CBOL Plant Working Group [26]

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
