# Peer review of "A Molecular Phylogenetic Study of the Genus Phedimus for Tracing the Origin of “Tottori Fujita” Cultivars"

_plants, 2020, doi:10.3390/plants9020254_

Round 1
Reviewer 1 Report
Review of plants-715067: A molecular phylogenetic study of the genus Phedimus for tracing the origin of “Tottori Fujita” cultivars
authors: Sung Kyung Han1 · Tae Hoon Kim2 · Jung Sung Kim1 5
reviewer Herve Seligmann
The manuscript is well written (but see edits listed) and apparently gives adequate background on the natural history of the plants (this should be developed (seasonality, habitats etc)) and their taxonomy/phylogeny.
The sequence material seems adequate but could be improved by using also maternally inherited sequence material (see below). The manuscript lacks discussing and developing discussion on:
1. hybridization
2. de novo mutations that are not clade specific
3. review literature on correlations and lack of correlations between molecular and morphological evolution/systematics.
The ms should propose to use also maternally inherited sequences (chloroplast and/or mitochondrial genomes) to test for possible directional hybridizations, where clade A might result from male gametes of clade B and female gametes of clade C etc, and adequately cite examples for this phenomenon. If the used sequence/s are maternally inherited, this must be stated and explained, and developed while evaluating results and differences between results for different genes.
The ms should describe other cases, from the literature, where such species complex occur, especially in cultivates vs wild plants.
These shortcomins should be relatively easily adressed by adding sections to the introduction and the discussion. After that, the ms should be publishable.
line
32 belongs->belong
48 tribus->tribes
49 subtribus->subtribes
84 plants -> plant
160 after correctING FOR the inversion
161 NumberS on branchES indicate bootstrap values...
247 the funded by the Ministy of Education -> sentence makes not sense
248 The authors express their appreciation
249 a wild location OR wild locations and a Japanese market/Japanese markets
Author Response
Dear editor and reviewers
Thank you for your kind advice for improving our manuscript. We checked again whole manuscript and revised it according to your comments as below.
Reviewer 1
The manuscript is well written (but see edits listed) and apparently gives adequate background on the natural history of the plants (this should be developed (seasonality, habitats etc)) and their taxonomy/phylogeny.
The sequence material seems adequate but could be improved by using also maternally inherited sequence material (see below). The manuscript lacks discussing and developing discussion on:
hybridization de novo mutations that are not clade specific review literature on correlations and lack of correlations between molecular and morphological evolution/systematics.
The ms should propose to use also maternally inherited sequences (chloroplast and/or mitochondrial genomes) to test for possible directional hybridizations, where clade A might result from male gametes of clade B and female gametes of clade C etc, and adequately cite examples for this phenomenon. If the used sequence/s are maternally inherited, this must be stated and explained, and developed while evaluating results and differences between results for different genes.
The ms should describe other cases, from the literature, where such species complex occur, especially in cultivates vs wild plants.
These shortcomins should be relatively easily adressed by adding sections to the introduction and the discussion. After that, the ms should be publishable.
→ We have revised the manuscript to expand the discussion section for explaining the discordance of the taxa in the tree and added the relevant references.
Most of all, we tried to discuss the hybridization event which could affect the diversification of the genus Phedimus considering the previous reports of hybrid origin in the wild species of the genus. And we found some example which are supposed to be originated by the hybridization from our result and provided the detail of them. And also, we discussed the probability of de novo mutation that caused the apparently different tree position.
For the concept of species complex, there was no study about that in the genus Phedimus up to dates. But we agree that it will be more reasonable to recognize as complex in some taxa. So we just explain the relationship of P. kamtschaticus and P. aizoon as the case in the discussion because their complexity was proved by molecular data in the present study.
line
32 belongs->belong
48 tribus->tribes
49 subtribus->subtribes
84 plants -> plant
160 after correctING FOR the inversion
161 NumberS on branchES indicate bootstrap values...
247 the funded by the Ministy of Education -> sentence makes not sense
248 The authors express their appreciation
249 a wild location OR wild locations and a Japanese market/Japanese markets
→ All the corrections have been completed in the revised manuscript.
Reviewer 2 Report
1). Manuscript ID: Plants-715067
2). Manuscript Title: A molecular phylogenetic study of the genus Phedimus for tracing the origin of “Tottori Fujita” cultivars
3). Comments:
The experiments are thoughtfully conceived and conducted. However, the following changes needs to be done before acceptance.
--Add scientific authority at the end of binomial names of all species when they are mentioned for the first time in the manuscript.
--Include full forms of all abbreviations/acronyms mentioned in the manuscript.
Lines 101 to 114: Include a figure to show sequence variation in psbA-trnH region.
In figures 3, 4 and 5 include full distance scale. It will be convenient for the readers to follow the information in Lines 116 to 135.
Lines 224 to 227: Maintain font uniformity.
Author Response
Dear editor and reviewers
Thank you for your kind advice for improving our manuscript. We checked again whole manuscript and revised it according to your comments as below.
Reviewer 2
The experiments are thoughtfully conceived and conducted. However, the following changes needs to be done before acceptance.
--Add scientific authority at the end of binomial names of all species when they are mentioned for the first time in the manuscript.
--Include full forms of all abbreviations/acronyms mentioned in the manuscript.
→ All the corrections about scientific authority and abbreviations/acronyms have been added in the revised manuscript.
Lines 101 to 114: Include a figure to show sequence variation in psbA-trnH region.
→ We added a supplementary figure including the inversion that widespread in the genus and a species-specific sequence variation found in P. takesimensis.
In figures 3, 4 and 5 include full distance scale. It will be convenient for the readers to follow the information in Lines 116 to 135.
→ The genetic distance between the Phedimus and outgroup was so far to show in a single page and it disturbs to understand the phylogenetic relationship among the Phedimus members. That is the reason why we make it shorten branch in the figure.
Lines 224 to 227: Maintain font uniformity.
→ Unmatched font format has been corrected in the revised manuscript.
Round 2
Reviewer 1 Report
The comments I made on the first version were clear and not difficult to implement. I do not think I need to reevaluate this manuscript.
Author Response
Dear Reviewer
Thank you very much for your interest in our manuscript and kind comment for improving it.
We could not find any specific point for further revision of our manuscript in your second comment so we would like to express again appreciation to your effort for our study.
Regards,
Jungsung